# EVIDENCE-ENHANCED TRIPLET GENERATION FRAME-WORK FOR HALLUCINATION ALLEVIATION IN GENERATIVE QUESTION ANSWERING

## ABSTRACT

To tackle the issue of hallucination in generative question answering (GQA)—where the generated answer is nonsensical or unfaithful to the provided document—we introduce a novel framework called evidence-enhanced triplet generation (EATQA). This framework incentivizes the model to generate all possible combinations of ⟨Question, Evidence, Answer⟩ triplets by reversing the source pair and target label to grasp their logical interrelationships. Specifically, the model predicts the Answer (A), Question (Q), and Evidence (E) given the QE, EA, and QA pairs, respectively. Furthermore, we address the distribution gap during the inference stage to extract knowledge from the evidence more effectively. Our framework ensures that the model comprehends the logical connections between queries, evidence, and answers, thereby simultaneously enhancing evidence generation and question answering capabilities. In this study, we apply the EATQA framework to the LLama model, demonstrating superior performance compared to other large language model (LLM)-based methods and hallucination mitigation techniques on two challenging GQA benchmarks. Further analysis reveals that our method not only preserves the pre-existing knowledge within the LLM but also reduces hallucination and produces more accurate answers.

## 1 INTRODUCTION

Large language models (LLMs) signify a pivotal advancement in the pursuit of general artificial intelligence (Brown et al., 2020; Touvron et al., 2023; Chowdhery et al., 2023). Despite their remarkable performance across a broad range of tasks, these models continue to encounter several challenges, such as hallucination (Tonmoy et al., 2024) and difficulties in processing long contexts (Jin et al., 2024). In the context of document-based generative question answering (GQA) (Lewis & Fan, 2018), models sometimes produce answers that are inconsistent with the source document or do not align with the query, a phenomenon known as hallucination (Gunjal et al., 2024; Liu et al., 2024). Recent studies have employed external models to retrieve pertinent information in an attempt to enhance the factual accuracy of generated responses. Nonetheless, the inherent mismatch between the retriever and the LLM can lead to the inclusion of superficially relevant information that does not contribute meaningfully to answering the question (Salemi & Zamani, 2024).

To enhance logical reasoning and minimize the inclusion of misleading information, we emphasize the identification of supporting evidence in document-based question answering (QA). Departing from the traditional retrieve-then-read approach, we employ a unified triplet generation framework where a large language model (LLM) simultaneously generates evidence and answers. Within this framework, pairs of ⟨question, evidence, answer⟩ are inputted into specific instructions to produce the remaining element. This approach leverages evidence to reconstruct the question, ensuring that the model grasps its logical relationships to both the question and the answer, rather than relying on superficial relevance.

Consider an example from the MultiRC dataset (Khashabi et al., 2018), illustrated in Figure 1. The question posed is, "After the Osprey resumed flights, how long did it take for the Air Force to begin using the aircraft?" The answer cannot be derived from a single sentence within the document. To accurately respond, the model must identify multiple pieces of evidence: "Osprey resumed flights

in 2002" and "Air Force began using Ospreys in 2008 after testing the aircraft in 2006" and then determine that the answer is "4 years" If the model is misled by incorrect evidence such as "Marines developed the aircraft in Iraq in 2007" it will arrive at the incorrect answer, "5 years" Moreover, when guided by the correct evidence, the model can accurately reconstruct the original question, since the correct evidence encompasses sufficient information. In contrast, incorrect evidence leads to the reconstruction of a question like "How long did it take for the Marines to begin using the aircraft..."-a question inconsistent with the original. This demonstrates that accurate evidence is vital for effective question answering, and the reconstruction of the question based on evidence and answer serves as an indicator of evidence validity.

Figure 1: One example from MultiRC dataset. Red denotes supporting evidence and green denotes misleading sentences.

To alleviate the hallucination and enhance the logical reasoning between the question, evidence and answers, we propose our **E**vidence enh**A**nced **T**riplet generation framework (EATQA), which includes three instruction tuning tasks to predict all the combinations of ⟨Question, Evidence, Answer⟩ triplet by flipping the source pair and the target label to understand their logical relationships, i.e., predict A(Answer), Q(Question), and E(Evidence) given a QE, EA, and QA pairs, respectively. We reduce the distribution gap between evidence-aware and evidence-absent QA settings through distribution bridging, thereby facilitating knowledge distillation from evidence and addressing challenges at the inference stage when evidence sentences cannot be explicitly derived.

We conduct experiments in a variety of widespread document-based GQA datasets with diverse answer types, including MultiRC and QASPER, based on different sizes of LLMs. Compared with different sizes of the backbone model, our unified triplet generation framework shows significant improvement on the two datasets, becoming the new state-of-the-art. Further analysis demonstrates the ability of our approach to tackle longer document with more sentences. Additionally, we observe a positive correlation in the performance of the three subtasks within the triplet generation framework, indicating the efficacy of unifying the generation of all components with a single LLM in this framework.

We conclude our contributions as follows: **1.** We highlight the evidence generation to alleviate hallucinations of LLM in GQA task. Instead of utilizing another LM as the retriever, which may introduce misleading information, we propose the unified evidence enhanced triplet generation framework including three instruction tuning tasks to improve the logical reasoning ability of LLM for GQA task. **2.** We propose the self-reasoning module, including the two phrase of candidate generation and correctness verify, which constructs the faithful and informative evidences for training without external annotation. **3.** We conduct experiments on a wide variety of multi-hop QA datasets including MultiRC and QASPER with different sizes of LLM, and demonstrate the effectiveness over existing methods. **4.** Additional experiments confirm the effectiveness of our unified triplet generation framework in both evidence retrieval and question answering. Furthermore, our method not only retains the prior knowledge encapsulated within the LLM but also effectively reduces hallucinations for questions that extend beyond the model's internal knowledge base.

## 2 RELATED WORK

Generative question answering (GQA) aims to generate an abstractive answer rather than extract an answer to a given question from provided passages (Fan et al., 2019; Li et al., 2021). Early works on GQA mostly tried to improve the faithfulness of the answer by investigating reliable external knowledge sources or incorporating multiple information sources. Yin et al. (2015) propose Neural

Generative Question Answering, an end-to-end model that generates answers to simple factoid questions based on the knowledge base, while Bi et al. (2019) propose the Knowledge-Enriched Answer Generator (KEAG) to generate a natural answer by integrating facts from four different information sources, namely, questions, passages, vocabulary, and knowledge.

Recent works focus more on the conditional generation model. Li et al. (2021) propose Rationale-Enriched Answer Generator (REAG), in which they add an extraction task to obtain the rationale for an answer at the encoding stage, and the decoder is expected to generate the answer based on both the extracted rationale and original input. Su et al. (2022) propose a framework named RBG (read before generate), to jointly models answer generation with machine reading. They augment the generation model with fine-grained, answer-related salient information predicted by the MRC module, to enhance answer faithfulness. Such methods can exploit and utilize the information in the original input better, while they require the extra effort of building models to extract that information. CAD (Shi et al., 2023) follows a contrastive output distribution that amplifies the difference between the output probabilities when a model is used with and without context. RHO (Ji et al., 2023) introduce local and global knowledge-grounding techniques into dialogue generation and further utilize a conversational reasoning model to re-rank the generated responses.

Our approach differs from these methods in 4 folds: **1**. The external information incorporated by existing baselines may be surface relevant but does not contain the information to support query answering, which introduces distraction for model generation. However our ability of generating informative evidences and conducting query reasoning improve as training proceeding. **2**. In existing baselines, the correctly exploit of external information beyond the internal knowledge to solve the query of model remains a challenge. However, our model needs to generate the evidence sentence from the document instead of internal knowledge, so it is trained to focus more on the document which mitigates hallucination. **3**. Our method does not need external pretrained retriever or well-designed knowledge base to mitigate the hallucination of backbone model. **4**. We provide the theory analysis to explain and demonstrate the effectiveness of our method design.

## 3 METHODOLOGY

In this section, we begin by introducing self-reasoning module to derive the faithful and informative evidences for training. Subsequently, we introduce the unified triplet generation framework designed to predict all possible combinations of ⟨Question, Evidence, Answer⟩ triplets by interchanging the source pair and target label to understand their logical interrelationships. These processes are illustrated in Figure 2, presented sequentially from top to bottom.

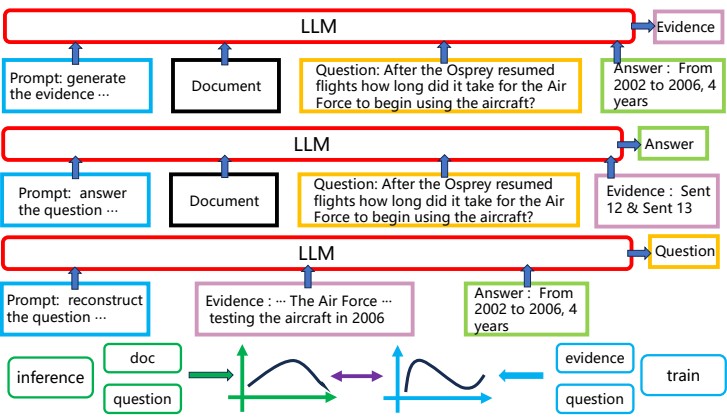

Figure 2: Model overview of EATQA.

The motivation behind the triplet generation framework is rooted in the idea that, according to Bayesian formulation:

$$\mathbb{P}(a|q,e,d) = \frac{\mathbb{P}(a,q,e,d)}{\mathbb{P}(q,e,d)} = \frac{\mathbb{P}(a,d)\mathbb{P}(e|a,d)\mathbb{P}(q|e,a,d)}{\mathbb{P}(q,e,d)} \tag{1}$$

where $d$, $q$, $e$, $a$ denote the document, question, evidence and answer. The posterior probability of accurately answering a question is positively proportional to the probability of generating evidence and reconstructing the question. This relationship suggests that enhancing evidence generation and question recovery can directly improve the reliability and accuracy of question answering. We assume the evidence sentences contain the sufficient information to reconstruct the question, i.e. $\mathbb{P}(q|e, a) = \mathbb{P}(q|e, a, d)$.

To establish the feasibility of our framework, we illustrate its functionality using query restoration as an example. In Figure 1, if the model is only provided with the answer "4 years" it faces difficulty in accurately reconstructing the query due to the potential presence of multiple sentences within the document that involve the phrase "4 years" However, when supplied with evidence sentences that highlight the key events, such as "Osprey resumed flights" and "Air Force began using the aircraft" the model can derive the essential components of the query. This enables our query restoration module to function effectively, thereby enhancing the model's ability to organize information and accurately reconstruct the query.

## 3.1 PRELIMINARY

The task of document-based generative question answering (GQA) involves producing an answer to a natural language question, relying on information from a document composed of multiple sentences. The model can be formulated as a function of

$$\mathbf{f_M}(\mathbf{a}) = \prod_{i=1}^{n_a} \mathbb{P}(a_i|a_0, a_1, a_2, \cdots, a_{i-1}, q, d) \tag{2}$$

where $n_a$ denotes the answer length, $q$ denotes the query, $d$ denotes the document including multiple sentences and $a_0$ denotes the begin-of-speech (BOS) token. Generally the answer has flexible forms which can not be directly extracted from the document.

## 3.2 SELF-REASONING

In the absence of annotated evidence within the GQA dataset, we adopt the principle that accurate evidence should fully encapsulate the information necessary to address the query independently of the document. Consequently, we employ the LLM to reason from its generated evidence. Specifically, we introduce a methodology termed self-reasoning, which involves two components: candidate generation and correctness verification.

During candidate generation, the LLM is instructed to produce candidate evidence supporting the query answering. This includes the original text from the document, while out-of-document candidates are filtered out to ensure the maintenance of factual accuracy. Though the filtered candidates are faithful, they do not necessarily contain the needed information for query (uninformative). In the correctness verification stage, the LLM provides a response to the query based on the initially generated candidates respectively. Evidence that fails to contain the required information will lead to incorrect answers. Therefore, we evaluate the predicted answer against the correct answer denoted as $a^*$, so as to eliminate evidence that may be factually accurate but lacks informative value:

$$e_i = M[p_e, d, q, s_i] \tag{3}$$
$$a_i = M[p_a, q, e_i] \tag{4}$$
$$e = \{e_i|a_i = a^*\} \tag{5}$$

where $s_i$ denotes the i-th random seed to sample for the evidence generation, $p_e$ denotes the prompt to generate evidence from the document to answer the query, $p_a$ denotes the prompt to generate the answer based on the query and evidence, and $e$ denotes the filterer evidences for further training. To this end, we construct the faithful and informative evidences for training without external tool.

## 3.3 TRIPLET GENERATION PARADIGM

Our triplet generation paradigm composes 3 modules, including Answer-Aware Evidence Generation (QAE), Evidence-Enhanced Question Answering (QEA), Evidence-Aware Question Restoration (EAQ). QAE enables the model to focus on the document, extracting critical information directly

from the text rather than relying on prior knowledge. QEA allows the model to leverage the available evidence effectively, ensuring that answers are grounded in the provided information and minimizing the risk of hallucination. EAQ facilitates the integration of evidence-derived information into the reasoning process, supporting more accurate and contextually relevant question restoration.

### 3.3.1 ANSWER-AWARE EVIDENCE GENERATION (QAE)

In this part, we model the probability of supporting evidence extraction for the query-answer pair $\mathbb{P}(e|a, q, d)$. We design a specific instruction for the LLM to generate evidence that supports both the query and the corresponding answer. Therefore, the input to model is the instruction, source document, the query and the corresponding answer. The output of model is the supporting evidence. The specific instruction is "generate the relevant evidence from the document to answer the following question" and we insert the document, question and answers into the template in Figure 5.

As for the loss function, by Bayesian Formula (Mises, 1942) we derive

$$
\begin{aligned}
\log(\mathbb{P}(e, q, d)) &= \log \int \mathbb{P}(e, q, a, d) d_a \\
&= \log \int \mathbb{Q}(a|e, q) \frac{\mathbb{P}(e, q, a, d)}{\mathbb{Q}(a|e, q)} d_a \\
&\geq \int \mathbb{Q}(a|e, q) \log(\frac{\mathbb{P}(e, q, a, d)}{\mathbb{Q}(a|e, q)}) d_a \\
&= E_{\mathbb{Q}(a|e,q)} \log(\frac{\mathbb{P}(e, q, a, d)}{\mathbb{Q}(a|e, q)}) \\
&= E_{\mathbb{Q}(a|e,q)} \log(\frac{\mathbb{P}(a, q, d)\mathbb{P}(e|a, q, d)}{\mathbb{Q}(a|e, q)}) \\
&= E_{\mathbb{Q}(a|e,q)} \log(\mathbb{P}(e|a, q, d)) + E_{\mathbb{Q}(a|e,q)} \log(\frac{\mathbb{P}(a, q, d)}{\mathbb{Q}(a|e, q)}) \\
&= E_{\mathbb{Q}(a|e,q)} \log(\mathbb{P}(e|a, q, d)) + E_{\mathbb{Q}(a|e,q)} \log(\frac{\mathbb{P}(a|q, d)}{\mathbb{Q}(a|e, q)}) + E_{\mathbb{Q}(a|e,q)} \log(\mathbb{P}(q, d)) \\
&= E_{\mathbb{Q}(a|e,q)} \log(\mathbb{P}(e|a, q, d)) - \mathbf{KL}(\mathbb{P}(a|q, d)||\mathbb{Q}(a|e, q)) + \log(\mathbb{P}(q, d))
\end{aligned}
\tag{6}
$$

where $\mathbb{Q}(a|e, q)$ denotes the probability of answer $a$ to the question $q$ holds based on the evidence $e$, which is produced by the same backbone in our method with specific prompt, $KL$ denotes Kullback-Leibler divergence (Van Erven & Harremos, 2014). To maximize the evidence extraction probability, we should maximize the probability of evidence supporting the question-answer pair $\mathbb{P}(e|a, q)$ and minimize the distribution distance between question answering with or without evidence $\mathbf{KL}(\mathbb{P}(a|q, d)||\mathbb{Q}(a|e, q))$. Considering the correct evidences contain identical information as the original document for the query reasoning, the second term $\mathbf{KL}(\mathbb{P}(a|q, d)||\mathbb{Q}(a|e, q))$, named as "**distribution bridging**", narrows down the gap between prediction based on the evidences and document, It enables LLM to make full use of evidences information to reason for answers. we utilize cross-entropy loss function to optimize the probability $\mathbb{P}(e|a, q)$:

$$
\mathcal{L}_{\text{QAE}} = -\log \mathbb{P}(e \mid d, q, a) = -\sum_{t=0}^{n_e-1} \log \mathbb{P}(e_{t+1} \mid d, q, a, e_{\leq t})
\tag{7}
$$

where $d$ denotes the document, $n_e$ denotes the length of the evidence, $\mathbb{P}(e_1 \mid d, q, a, e_{\leq 0}) := \mathbb{P}(e_1 \mid d, q, a)$.

### 3.3.2 EVIDENCE-ENHANCED QUESTION ANSWERING (QEA)

In this part, we task LLM with generating answers based on the corresponding question and the relevant evidence. The instruction provided is "generate the correct answers for the following question based on the document and the evidence support the answers to the question", and we incorporate the instruction, document, question and evidence into the template in Figure 5, as inputs into the LLM. The objective function formulated as:

$$
\mathcal{L}_{\text{seq}} = -\log \mathbb{P}(a \mid d, q, e) = -\sum_{t=0}^{n_a-1} \log \mathbb{P}(a_{t+1} \mid d, q, e, a_{\leq t})
\tag{8}
$$

where $n_a$ denotes the length of the answers, $\mathbb{P}\left(a_1 \mid d, q, e, a_{\leq 0}\right) := \mathbb{P}\left(a_1 \mid d, q, e\right)$. This task can be seen as the main task of EATQA and enables the model to derive the answers based on the question and evidence. On the other hand, to narrow the gap between training and inference, we minimize the second term of Eq.6: $\mathbf{KL}(\mathbb{P}(a|d,q)||\mathbb{Q}(a|e,q))$. When the evidences are incomplete or have misleading information, the model resorts to the original document for the answer, which improves the robustness of training stage. Therefore, the loss function of this part is:

$$\mathcal{L}_{\text{QEA}} = \mathcal{L}_{\text{seq}} + \alpha_{kl} \cdot \mathbf{KL}(\mathbb{P}(a|d,q)||\mathbb{Q}(a|e,q)) \tag{9}$$

where $\alpha_{kl}$ denotes the hyper-parameter to tune.

### 3.3.3 Evidence-Aware Question Restoration (EAQ)

In this part, we aim to model the probability of $\mathbb{P}(q|e,a)$ and instruct the LLM to recover the question based on the evidence-answer pair. The prompt given is "reconstruct the question based on the answers and corresponding supporting evidence", and we integrate the prompt, document, evidence and answers into the template in Figure 5. The objective function is formulated as:

$$\mathcal{L}_{\text{EAQ}} = -\log \mathbb{P}(q \mid d, e, a) = -\sum_{t=0}^{n_q-1} \log \mathbb{P}\left(q_{t+1} \mid d, e, a, q_{\leq t}\right) \tag{10}$$

where $n_q$ denotes the length of the question, $\mathbb{P}\left(q_1 \mid d, e, a, q_{\leq 0}\right) := \mathbb{P}\left(q_1 \mid d, e, a\right)$. Considering the incorrect evidence does not contain the full information of the original question, this objective helps to enhance the casual relations between evidence and answers.

### 3.4 Training and Inference

With the unified optimization of all three EATQA objectives, our model captures the logical relations between question, evidence and answers. Based on the probability induction:

$$\log \mathbb{P}(a|q, e, d) \propto \log(\mathbb{P}(a|d, q)) + \log(\mathbb{P}(e|a, d)) + \log(\mathbb{P}(q|e, a, d))$$

The overall objective is the weighted accumulation:

$$\mathcal{L}_{Triplet} = \alpha_1 \mathcal{L}_{QAE} + \alpha_2 \mathcal{L}_{QEA} + \alpha_3 \mathcal{L}_{EAQ} \tag{11}$$

where $\alpha_1$, $\alpha_2$ and $\alpha_3$ are tuneable hyper-parameters.

Because of the design of distribution bridging, we do not need to first generate the evidence based on the question and then construct QEA template to Table 5. Instead, we can directly instruct the model to generate the answer from the original document, which keeps the inference efficiency.

## 4 Experiments

### 4.1 Datasets

We evaluate on a diverse variety of widespread benchmark multi-hop QA datasets, including MultiRC (Khashabi et al., 2018), QASPER (Dasigi et al., 2021), NQ Kwiatkowski et al. (2019), HotpotQA Yang et al. (2018), TriviaQA Joshi et al. (2017), StrategyQA Geva et al. (2021) across different domains. We utilize Exact Match (EM) and F1 scores (Opitz & Burst, 2019) to evaluate our method. The F1 score measures the overlap of answer tokens between the predicted and ground-truth answer. EM is more strict which awards point if any of the annotated answers is generated exactly.

### 4.2 Implementation Details

We conduct experiments with LLama2 (Touvron et al., 2023) from 7B to 13B as the LLM. To reduce computation cost and keep prior knowledge in LLM, we use LoRA (Hu et al., 2021), which freezes the pretrained model weights and injects trainable rank decomposition matrices into each layer of the LLM. We tune the parameters based on the develop set and the parameters $\alpha_1, \alpha_2, \alpha_3$ in Eq. 11 and $\alpha kl$ in Eq. 9 are tuned from [0.1, 0.3, 0.5, 0.7, 1.0], and set to 0.3, 1.0, 0.3 and 0.5 in our method. We use AdamW as optimizer and the initial learning rate is set to 3e-5. GPT-3 reports few shot results with 32 examples in the prompt without parameter updating. Because the maximum input length of LLama2 is 4096 and the average context length of QASPER is about 16K, we utilize position interpolation (Chen et al., 2023) to extend the context length to 32K.

## 4.3 BASELINES

We compare our method with existing widespread LLMs including T5-11B (Raffel et al., 2020), Flan-137B (Wei et al., 2021), Vega2-6B (Zhong et al., 2022), GPT-3 (few shot) (Brown et al., 2020), LoRAMoE (Dou et al., 2023), PaLM 540B (Anil et al., 2023) for MultiRC. For QASPER, we compare our method with LLM-based long context methods, AttenWalker (Nie et al., 2023), ChatGLM3-6B-32k (Du et al., 2021), SE-Mistral-7B Jin et al. (2024), VCC-3B (Zeng et al., 2024) and TOVA-7B (Oren et al., 2024) For hallucination mitigation methods, we compare our approach against RAG Lewis et al. (2020) with Dense Passage Retriever (DPR) (Karpukhin et al., 2020), CAD (Shi et al., 2023), RHO (Ji et al., 2023) using the same backbone. These 3 methods are representative methods of 3 different categories of hallucination mitigation: retrieval Augmented Generation, Introducing New Decoding Strategy, and Utilization of Knowledge Graph (KG). In Table 1, CAD and RHO results are reproduced with the code provided in original paper using the same backbone with ours for fair comparison.

## 4.4 EFFECTIVE TRIPLET GENERATION

From Table 1, compared with the backbone, our method improves by 4.6 EM and 2.4 F1 on 7B-scale model as well as 3.6 EM and 1.9F1 on 13B-scale model. It demonstrates the effectiveness of our evidence enhanced triplet generation framework on document based GQA. Moreover, our method with 13B parameters outperforms the 540B PaLM finetuning by 2.0 EM and 1.1 F1, becoming the new state-of-the-art. Our method with 7B-scale model has achieved the comparable performance on F1 with larger models like T5-xxl and Flan-T5.

From Table 1 compared with the backbone, our method improves by 3.0 F1 on 7B-scale model. QASPER contains more rigorous samples and existing hallucination mitigation methods struggle to improve the performance. It demonstrates the effectiveness of our method on challenging long document QA.

| Methods | MultiRC | | QASPER | #Para. |
|---|---|---|---|---|
| | EM | F1 | F1 | |
| GPT-3 (32 shot) | 30.5 | 75.4 | - | 175B |
| Flan-T5 | - | 83.4 | - | 137B |
| T5 | 63.1 | 88.1 | - | 11B |
| ERNIE-3.0 | 63.2 | 88.6 | - | 10B |
| PALM | 63.6 | 88.7 | - | 540B |
| SE-Mistral-7B | - | - | 39.3 | 7B |
| TOVA-7B | - | - | 42.0 | 7B |
| ChatGLM3-6B-32k | - | - | 43.3 | 6B |
| LLama2-7B | 57.2 | 86.1 | 42.4 | 7B |
| RAG | 58.1 | 86.7 | 43.9 | 7B |
| CAD | 58.2 | 87.2 | 43.1 | 7B |
| RHO | 59.4 | 87.3 | 43.2 | 7B |
| EAT-QA-7B | 61.8 | 88.5 | 45.4 | 7B |
| LLama2-13B | 62.0 | 87.9 | 45.1 | 13B |
| RAG | 63.1 | 88.1 | 44.9 | 13B |
| CAD | 63.5 | 88.3 | 45.8 | 13B |
| RHO | 64.2 | 88.4 | 45.9 | 13B |
| EATQA-13B | **65.6** | **89.8** | **48.1** | 13B |

Table 1: Results on MultiRC and QASPER dataset compared with competitive LLM methods. "#Para." denotes the parameter number in the model. We conduct 5 experiments with different random seeds and our method significantly beats the prior SOTA, with p-value less than 0.001.

# 5 ABLATION, GENERALIZATION AND HALLUCINATION MITIGATION

## 5.1 ABLATION

In this part, we investigate the effectiveness of different modules in our method, including QAE, EAQ and the distribution bridging.

**Does question restoration matter?** In this ablation, we remove the module of question restoration and investigate its effect on question answering. In table 2, removing question restoration will drop 1.6 EM and 1.4 F1 with 7B model, as well as 1.4 EM and 1.1 F1 with 13B model. Considering the context is not inputted into model in the query restoration module, the model has to utilize the information in evidence to recover the question. This module enhances the ability to integrate multiple pieces of information in evidence sentences, and understand logical relation between query, answer and evidence for LLM, which shows the effectiveness for GQA.

| Probability | LLama2 | EATQA |
|---|---|---|
| $\mathbb{P}(Y_{A\|Q} = \hat{Y})$ | 34.8 | 37.1 |
| $\mathbb{P}(Y_{A\|Q,D} = \hat{Y}\|Y_{A\|Q} = \hat{Y})$ | 88.8 | 85.8 |
| $\mathbb{P}(Y_{A\|Q,D} = \hat{Y}\|Y_{A\|Q} \neq \hat{Y})$ | 48.7 | 52.2 |

Table 3: Prior knowledge mitigation and hallucination mitigation. $Y_{A|Q}$ denotes the answer generated based on the vanilla query by QA model, which reflects the prior knowledge of LLM. $Y_{A|Q,D}$ denotes the answer generated based on the query and document. $\hat{Y}$ denotes the golden answer.

**Does evidence generation matter?** In this ablation, we remove the module of evidence generation and investigate its effect on GQA. In Table 2, removing evidence restoration will drop 1.0 EM and 0.8 F1 with 7B model, as well as 1.1 EM and 1.2 F1 with 13B model. Evidence extraction encourages the model to reason for the supporting facts that entail the question-answer pair, which enhances the understanding of logical relation among query, answer and evidence. Removing evidence generation decreases the attention of model pays to the important facts in the document.

**Should we narrow down the distance between** $\mathbb{P}(a|dq)$ **and** $q(a|e, q)$**?** In this ablation, we remove the KL-divergence loss in Eq.6 in training. In inference stage, we input the predicted evidence and the query to derive the answer. In Table 2, removing KL loss will drop 0.8 EM and 0.9 F1 with 7B model, as well as 1.0 EM and 0.7 F1 with 13B model. Though keeping effective performance, the distribution bridging distills the knowledge of evidence and narrows down the gap between training and inference, avoiding first retrieving the evidence and then inputting the evidence alongside the query into model to reason for the answer.

## 5.2 DIFFERENT DOCUMENT LENGTHS AND SENTENCE NUMBER

In this part, we assess our performance on cases with varying document lengths and sentence numbers comparing with the backbone. For this purpose, we divide the MultiRC development set into 4 distinct groups, categorized based on the document length and sentence number respectively, and apply F1 to evaluate the performance of different models. Groups are indexed by the ascending order of document length, i.e., Group 1 denotes cases in the percentile interval 0-0.25 of the full dataset and Group 4 denotes cases in the percentile interval 0.75-1.0. Therefore, groups 3 and 4 have longer documents than groups 1 and 2.

| Methods | EM | F1 | #Para. |
|---|---|---|---|
| w/ LLama2-7B | | | |
| backbone | 57.2 | 86.1 | 7B |
| -Question Restoration | 60.2 | 87.1 | 7B |
| -Evidence Generation | 60.8 | 87.7 | 7B |
| -KL | 61.0 | 87.6 | 7B |
| EATQA-7B | 61.8 | 88.5 | 7B |
| w/ LLama2-13B | | | |
| backbone | 62.0 | 87.9 | 13B |
| -Query Restoration | 64.2 | 88.7 | 13B |
| -Evidence Generation | 64.5 | 88.6 | 13B |
| -KL | 64.6 | 89.1 | 13B |
| EATQA-13B | 65.6 | 89.8 | 13B |

Table 2: Ablation results with LLama2 from 7B to 13B on MultiRC dataset.

Generally, our model derives significant improvement over LLama2-13B in groups with different document lengths and sentence numbers. It demonstrates the effectiveness of our evidence enhanced triplet generation framework on document-based GQA. In Table 4, EATQA outperforms LLama2 by 3.5 and 1.5 F1 in groups 3 and 4, as well as 1.8 and 1.2 F1 in groups 1 and 2. In Table 7, EATQA outperforms LLama2 by 3.4 and 2.7 F1 in groups 3 and 4. Longer context brings the difficulty for model to capture important information about the query and derive the correct answer. Our method enhances the capture of supporting information from the document, which mitigates the hallucination about distracting information.

## 5.3 PERFORMANCE ON EVIDENCE GENERATION

Not only deriving effectiveness on GQA, our method also shows improvement on evidence generation. In Table 5, comparing with sequentially generating evidence and answer, our method outperforms by 3.1 on 7B and 2.5 F1 on 13B. Considering our method first generates the evidence and integrates the information of evidence for answers, the evidences serve as the basis of reasoning process.

$$\mathbb{P}(a|q, e, d) \propto \mathbb{P}(e|a, d)\mathbb{P}(q|e, a, d)$$

| Group | 1 | 2 | 3 | 4 |
|---|---|---|---|---|
| Length | 379 | 486 | 587 | 726 |
| LLama2 | 88.3 | 90.7 | 82.9 | 87.8 |
| EATQA | 90.5 | 91.9 | 86.4 | 89.3 |

Table 4: Results on MultiRC dataset grouped by different document lengths. Groups are indexed by the ascending order of document length, i.e., Group 1 denotes cases in the percentile interval 0-0.25 of the full dataset. "length" denotes the average document length in the specific percentile interval and we utilize F1 to evaluate the model performance.

| Model | 7B | 13B |
|---|---|---|
| LLama2 | 59.8 | 62.7 |
| Joint decoding | 60.3 | 63.1 |
| EATQA | **63.4** | **65.6** |

Table 5: Performance on evidence generation in MultiRC dataset. We utilize token-level F1 score as the evaluation metric. "LLama" denotes instructing the LLM to generate the evidence only. "Joint Decoding" denotes sequentially generating evidence and answer.

Fixing the ability of information integration, the evaluation of evidences shows the ability of capturing key information beyond the distracting contents of the document so that generating faithful and correct answer instead of hallucination. Therefore, we demonstrate our evidence enhanced triplet generation paradigm significantly improves the ability of hallucination mitigation.

## 5.4 HALLUCINATION MITIGATION

Considering the prior knowledge within LLM, we observe for some "already-known" questions, the model can generate the correct answer without the document, such as "What is gravity's role in space?". We utilize $\mathbb{P}(Y_{A|Q} = \hat{Y})$ to evaluate the internal knowledge of model. When the model can not generate the correct answer without the document, the model resorts to the document rather than internal knowledge. The probability $\mathbb{P}(Y_{A|Q,D} = \hat{Y}|Y_{A|Q} \neq \hat{Y})$ denote that the model rely on the document to give the faithful answer beyond the incorrect internal knowledge, which can be utilized to evaluate the ability of hallucination mitigation (Qiu et al., 2023). In Table 3, our model significantly mitigates the hallucination while keeping prior knowledge to solve the "already-known" questions. In Tabl 8, we utilize GPT-4 to evaluate the hallucination rate of evidence generated and reasoning result, which also demonstrates our effectiveness over hallucination mitigation.

## 5.5 CORRELATION BETWEEN DIFFERENT MODULES

In this part, we explore the correlation of model performance in query answering (QEA), evidence generation (QAE) and query restoration (EAQ) on data samples. To mitigate the bias of extreme sample, we classify the samples in development set into 50 groups with same size based on the QEA F1. We take the average F1 score of all samples in the group as its overall F1 score. We respectively draw the scatter plot of each pair of QEA, QAE, EAQ score versus the other and fit with linear function. In Figure 3. we find the QAE score and EAQ score are directly proportional to QEA score. In our triplet generation framework, with better performance in evidence generation and query restoration, the model derives better performance in query answering. This shows the effectiveness of our EATQA, which enhances the understanding of LLM about logical relations between query, evidence and answer.

## 5.6 GENERALIZATION ON DIVERSE DATASETS

Following REACT (Yao et al., 2022), we utilize 2000 samples as the training set. In Table 6, our method derives significant improvement over existing hallucination mitigation methods on diverse multi-hop QA datasets.

## 5.7 ATTENTION WEIGHTS

In this part, we compute the average attention weights about query to document and evidence in the query answering task in respective layers of the transformer block. We conduct statistics on the development set of the MultiRC dataset with 13B model. In evidence-aware query answering, the model assigns about twice as much as attention weights to evidence token than context token. It shows the evidence contains denser information to derive the answer. Our implementation of

| Model | NQ | HotpotQA | TriviaQA | StrategyQA |
|-------|------|----------|----------|------------|
| Llama2 | 45.5 | 41.3 | 69.6 | 62.4 |
| RAG | 46.3 | 42.1 | 70.3 | 62.9 |
| REACT | 46.8 | 43.2 | 70.7 | 64.1 |
| CAD | 47.2 | 43.1 | 70.5 | 64.0 |
| RHO | 47.6 | 42.9 | 71.1 | 63.8 |
| EATQA | 49.1 | 44.9 | 73.4 | 65.2 |

Table 6: Performance on diverse datasets. We utilize F1 to evaluate NQ, HotpotQA and TriviaQA, and use accuracy to evaluate StrategyQA.

distribution bridging distills the abundant information in evidence to evidence-absent query answering in inference phrase. In the EAQ, the token-average attention weights of generated query paid to evidence are comparable to answer texts. Considering the evidence contains more tokens than the answer, this finding underscores the crucial role that evidence plays in the feasibility of the EAQ task.

## 5.8 COMPUTATION COST.

Considering the length of evidences is much less than the document (about 10% of the document length), and the transformer computation cost are quadratic relation to the input length, our evidence enhanced triplet generation paradigm will not significantly increase the computation cost. In practice, the baseline llama2 finetuning costs about 5 hours and our method costs about 7 hours with one A100 gpu. Considering our significant improvement over informative evidence generation as well as faithful answer reasoning, it shows the effectiveness of our evidence enhanced triplet generation paradigm. In inference stage, our method needs no more computation cost compared with vanilla Llama finetuning.

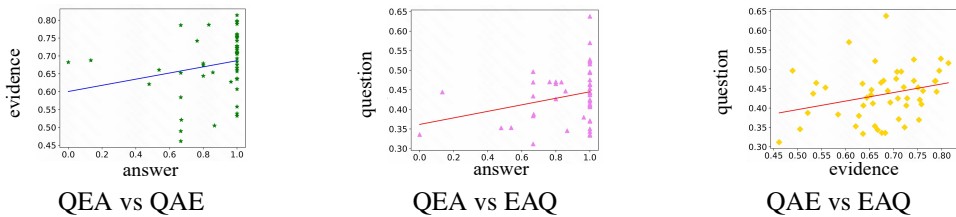

| QEA vs QAE | QEA vs EAQ | QAE vs EAQ |

Figure 3: Performance relevance between 3 modules in our method with 13B backbone. QEA denotes evidence-aware question answering, EAQ denotes evidence-grounded query restoration and QAE denotes answer-aware evidence retrieval.

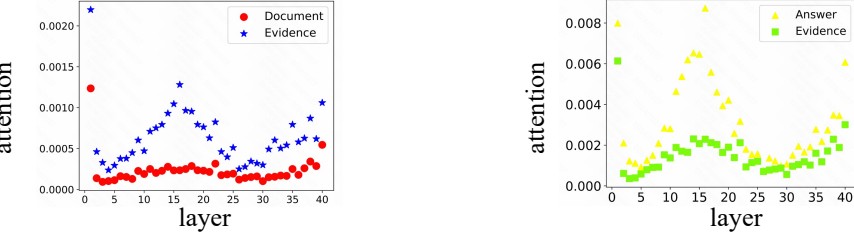

Figure 4: Attention weights about different layers with 13B backbone. The left graph denotes the attention weights of query to document and evidence in Evidence-Enhanced Question Answering stage; the right denotes the attention weights of generated query to evidence and answer in Evidence-Aware Question Restoration stage.

## 6 CONCLUSION

In this paper, we propose the unified triplet generation framework including three instruction tuning tasks to improve the logical reasoning ability of LLM for GQA task. We conduct experiments on a variety of widespread document-based QA datasets with different sizes of LLM, and outperform existing hallucination mitigation methods.

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

# A APPENDIX

**1. Input templates of different modules in EATQA.** We experiment multiple prompts and choose the optimal.

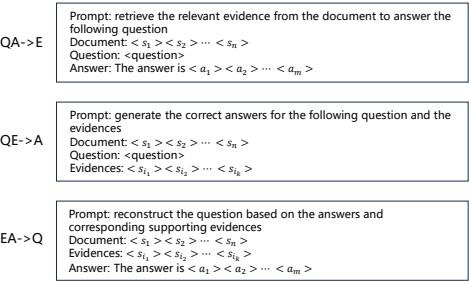

Figure 5: Input templates of EATQA.

**2. Results on MultiRC dataset grouped by different sentence numbers in the document.**

| Group  | 1    | 2    | 3    | 4    |
|--------|------|------|------|------|
| number | 10.8 | 13.5 | 16.0 | 18.3 |
| LLama2 | 87.5 | 85.1 | 85.8 | 89.0 |
| EATQA  | 90.7 | 84.3 | 89.2 | 91.7 |

Table 7: Results on MultiRC dataset grouped by different sentence numbers in the document. Groups are indexed by the ascending order of sentence number. "number" denotes the average sentence number in the specific percentile interval. We utilize F1 to evaluate the model performance.

| model              | Llama2 | RAG  | CAD  | RHO  | EATQA |
|--------------------|--------|------|------|------|-------|
| hal-rate $\downarrow$ | 27.5   | 24.3 | 25.6 | 22.8 | 17.2  |

Table 8: Evaluation results with GPT-4.

**3. Hallucination evaluation with external tool.**  To more comprehensively demonstrate our ability of hallucination mitigation. We follow Lei et al. (2023) to utilize GPT-4 to act as an external judge. We append the generated evidence and reasoning result as the input and prompt GPT-4 to evaluate the hallucination rate against the document and query on MuitiRC dataset. Based on the above result, our method significantly outperforms the existing baselines in decreasing the hallucination rate. In our triplet generation paradigm, considering the evidences are included in the document, our model relies on the document to derive supporting information instead of internal prior knowledge in the evidence generation module. Moreover, the "distribution bridging" module enables our model to make faithful prediction based on the informative evidences beyond other distracting contents in the document. In general, our model focuses on the faithful and informative evidences to conduct the reasoning process, which mitigates the hallucination.

**4. Dataset statistics.**  MultiRC creates multi-domain multi-hop questions, where documents across various domains are selected from multiple datasets. Each instance consists of a document including about 15 sentences. All instances were constructed such that it is not possible to answer a question correctly without gathering information from multiple sentences. QASPER includes 5049 questions over 1585 Natural Language Processing papers in the academic research domain focusing on entire papers, which is designed to facilitate document-grounded, information-seeking QA. QASPER contains a variety of answer types, including extractive, abstractive, yes/no, and unanswerable questions.

**5. Model Architecture.**  EATQA is built on the widespread LLM, Llama (Touvron et al., 2023) with a few additional learnable parameters. we additionally adopt several trainable adapter tokens $p = [p_1, p_2, \cdots, p_{N_p}]$ which are prepended to the key and value of each self-attention layer, where $N_p$ is the number of adapter tokens. So the number of trainable parameters of EATQA 7B is 4.5M, only 0.06% of total parameters of LLama 7B. With such a few trainable parameters, EATQA effectively preserves LLMs' prior knowledge and the casual reasoning ability to understand the logical relations between the question, evidence and answer. EATQA consists of three objectives: answer-aware evidence generation, evidence-enhanced query answering and evidence-aware query restoration.

