# OpenReview forum: "Evidence-Enhanced Triplet Generation Framework for Hallucination Alleviation in Generative Question Answering"
_ICLR.cc/2025/Conference — Submitted to ICLR 2025_

### Official Review · Reviewer_sUXJ · 2024-11-02

**Soundness:** 2
**Presentation:** 2
**Contribution:** 3
**Rating:** 5
**Confidence:** 3

**Summary:**

The paper proposed an evidence-enhanced triplet generation framework, EATQA, to address a hallucination issue in generative question answering (GQA). The EATQA encourages the model to predict Answer (A), Question (Q), and Evidence (E), given QE, EA, and QA pairs, respectively. that is, all the combinations of ⟨Question, Evidence, Answer⟩. to understand their relationships. The paper applied it to LLama, that outperformed other LLM-based methods and hallucination mitigation approaches on two GQA benchmarks.

**Strengths:**

The proposed triplet generation framework showed significant improvement on two widespread document-based GQA datasets, MultiRC and QASPER, yielding state-of-the-art performance on the datasets.

**Weaknesses:**

1. First, the paper was not necessarily written in good English. It should receive a native check. Further, it is partly difficult to understand. The authors incorrectly used LaTeX cite commands, that makes the draft more difficult to read. It is better to check the whole draft more carefully again.

2.  While the proposed framework could yield better performance in GQA tasks, the evaluation in hallucination alleviation was not necessarily thorough enough, that makes it difficult to judge whether the proposed framework is really good in the hallucination alleviation.  The analysis in Sec. 5.4 did not necessarily directly evaluate the degree of hallucination alleviation. Furthermore, no comparisons with previous related work were shown. It is better to show how well the proposed framework can alleviate hallucination directly and clearly, in comparison with related work.

3. In the analysis in Sec. 5.3, no explanation was provided for the performance in Table 6. If it is the evaluation for generated evidences, how reference evidences can be obtained because it was mentioned that evidence annotation is unavailable in the datasets? it is also not described how the scores were calculated.

4. The analysis in Sec. 5.2 seems to contribute to fewer useful findings. In my understanding, since the document length is proportional to the number of sentences, just a table might be enough from Tables 4 and 5.

5. It is better to clearly describe how the authors fixed hyperparameters in the experiments.

**Questions:**

1. What was the value for a hyperparameter \alpha_{kl} and how did the authors fix it?

---

> ### Author Response · Authors · 2024-11-23
> **Response for reviewer sUXJ**
>
> Thank you very much for providing such insightful and valuable suggestions. We have carefully considered the questions you proposed and would like to respond to each point in detail.
>
> # As for evaluation in hallucination alleviation
> 1. We need to clarify that the hallucination in GQA is defined as the generated content is nonsensical or unfaithful to a reference content [1]. One of the most common reasons of hallucination **comes from the over confidence of internal knowledge [2]**. So we follow existing method [3] to take the estimation of the posterior and prior probabilities of generated responses conditioned (or not conditioned, respectively) on the source document as a metric of hallucination mitigation. The **probability of  $ P(Y_{A|Q,D} = \hat{Y} | Y_{A|Q} \neq  \hat{Y})$ denotes the model can rely on the document to give the faithful answer beyond the incorrect internal knowledge**. So it **can be utilized to evaluate the ability of hallucination mitigation**.
> 2. We also compare the evidence generated in our method with baselines in Table 6. Considering our method first generates the evidence and integrates the information by evidence for answers, the evidences are the basis of reasoning process.
> $$ \mathbf{P_M}(a |q,e,d)  \propto \mathbf{P_M}(e|a,d)  \mathbf{P_M}(q|e,a,d)$$
> Fixing the ability of information integration, the evaluation of evidences shows the **ability of capturing key information beyond the distracting contents of the document so that generating faithful and correct answer instead of hallucination**. From Table 6, we demonstrate our evidence enhanced triplet generation paradigm significantly improves the ability of hallucination mitigation.
> 3. To more comprehensively demonstrate our ability of hallucination mitigation. We follow [4] to utilize GPT-4 to act as an external judge. We append the generated evidence and reasoning result as the input and prompt GPT-4 to evaluate the hallucination rate against the document and query on MuitiRC dataset.
> | model | Llama2  | RAG | CAD | RHO| EATQA |
> |  ----  | ----  |----  | ----  | ----  |----  |
> | hal-rate $\downarrow$|27.5 | 24.3 | 25.6 |22.8| 17.2|
>
> Based on the above result, our method significantly outperforms the existing baselines in decreasing the hallucination rate. In our triplet generation paradigm, considering the evidences are included in the document, our model **relies on the document to derive supporting information instead of internal prior knowledge** in the evidence generation module. Moreover, the **“distribution bridging” module enables our model to make faithful prediction based on the informative evidences beyond other distracting contents in the document** . In general, our model focuses on the faithful and informative evidences to conduct the reasoning process, which mitigates the hallucination.
> In conclusion, from **the above three well designed metrics, we demonstrate our ability of hallucination mitigation**.
>
> # As for the reference evidence
> We need to clarify that our method **does not need any annotated evidences thanks to our self-reasoning module**. We propose the self-reasoning method in section 3.2, which composes **candidate generation and correctness verify.** In candidate generation, the LLM is instructed to generate the candidate evidence including only the original text from document and out-of-document candidates are filtered to maintain the factuality. In correctness verify, LLM needs to answer the query based on the vanilla generated candidates. The evidences which did not contain the needed information will induce incorrect answers, so we compare the predicted answer against the golden answer to filter the factual faithful but not informative evidences. Through our self-reasoning module, we **avoid the use of external annotation tool to derive the faithful and informative evidences for training**. In evidence evaluation, we utilize token-level F1 to evaluate the predicted evidence against the derived evidence by self-reasoning. The improvement of evidence generation induces the more faithful answer generation, which demonstrates our effectiveness in hallucination mitigation. Actually, from Figure 4, our ability of **informative evidence generation and faithful query answering are improving at the same time**.
>
>
> [1] EVER: Mitigating Hallucination in Large Language Models through Real-Time Verification and Rectification. Arxiv 2023.
>
> [2] Mitigating Overconfidence in Large Language Models: A Behavioral Lens on Confidence Estimation and Calibration. NIPS 2024.
>
> [3] Detecting and Mitigating Hallucinations in Multilingual Summarization. EMNLP 2023.
>
> [4] Chain of Natural Language Inference for Reducing Large Language Model Ungrounded Hallucinations. Arxiv 2023.
>
> [5] A robust Spearman correlation coefficient permutation test. Communications in Statistics - Theory and Method 2024.
>
> [6] Think While You Write: Hypothesis Verification Promotes Faithful Knowledge-to-Text Generation. NAACL 2024.

---

> > ### Comment · Reviewer_sUXJ · 2024-12-01
> >
> > Thanks for the detailed response. I decide to keep my rating unchanged.

---

> ### Author Response · Authors · 2024-11-23
> **Response for reviewer sUXJ**
>
> # As for the  Sec. 5.2  information
> Yes, we **agree that the document length has positive correlation to the number of sentences**. However, one sentence describes a unit of semantic information and the longer document **does not necessarily mean more sentences**. In fact, the **Spearman correlation coefficient [5] between the document length and the sentence number in the develop set of MultiRC is only 0.29**, which is far from the complete association 1.0. It means the longer document **does not necessarily mean more sentences**.  So our original intention of this part is to investigate our performance given different types of documents in **a more comprehensive view (longer or more semantic units)**.  From Table 5. we can see our method is **effective across different number of sentences**. We will put this part in appendix as your valuable suggestions.
>
> # As for the hyperparameters in the experiments
> We follow existing methods [3,6] to fix the hypermeters based on the performance of develop dataset and report the results on the test dataset. The $\alpha_{kl}$ is set to 0.5.

---

> ### Author Response · Authors · 2024-11-30
> **Hope for Reply**
>
> Dear Reviewer sUXJ:
>
> We sincerely thank you for your valuable and constructive feedback. We have dedicated considerable time to crafting this rebuttal, as well as updated our paper with the revisions and additional experimental results in the revised PDF. We are sincerely willing to address any concerns you have and hope for your replies.
>
> Best regards.
>
> Authors.

---

### Official Review · Reviewer_7W21 · 2024-11-04

**Soundness:** 3
**Presentation:** 3
**Contribution:** 2
**Rating:** 6
**Confidence:** 3

**Summary:**

This paper proposes EATQA to address hallucination issues in GQA. It is an unified triplet generation approach that can capture logical relationships between question, evidence, and answer.

**Strengths:**

The method is well-motivated and the paper is easy to follow. The experiments show the proposed method has great improvements.

**Weaknesses:**

1. The method is based on gold evidence annotations when training. It may limit its applicability to datasets without such annotations.

2. The improvement margins on some baselines, e.g., CAD and RHO, are relatively modest.

3. Is the computational costs and inference time comparison to baselines missing?

**Questions:**

How does the method perform on datasets without gold evidence annotations?

---

> ### Author Response · Authors · 2024-11-23
> **Response for reviewer 7W21**
>
> Thank you very much for providing such insightful and valuable suggestions. We have carefully considered the questions you proposed and would like to respond to each point in detail.
>
> # As for gold evidence annotations
> We need to clarify that our method **does not need any annotated evidences thanks to our self-reasoning module**. We propose the self-reasoning method in section 3.2, which composes **candidate generation and correctness verify.** In candidate generation, the LLM is instructed to generate the candidate evidence including only the original text from document and out-of-document candidates are filtered to maintain the factuality. In correctness verify, LLM needs to answer the query based on the vanilla generated candidates. The evidences which did not contain the needed information will induce incorrect answers, so we compare the predicted answer against the golden answer to filter the factual faithful but not informative evidences. Through our self-reasoning module, we **avoid the use of external annotation tool to derive the faithful and informative evidences for training**. In evidence evaluation, we utilize token-level F1 to evaluate the predicted evidence against the derived evidence by self-reasoning. The improvement of evidence generation induces the more faithful answer generation, which demonstrates our effectiveness in hallucination mitigation. Actually, from Figure 4, our ability of informative evidence generation and faithful query answering are improving at the same time.
>
>
> # As for improvement
> We need to clarify that the two benchmarks we used are challenging datasets which involve multi-hop reasoning where answers can not be derived from one part of the document. **The existing baselines improve much less beyond the backbone llama2  (less than 1.0 F1 score on Qasper dataset) compared with ours (about 3.0 F1 score Qasper dataset). So our improvement over backbone Llama2 is not marginal**. We also conduct experiments on a diverse range of datasets as follows:
>
> | model | NQ  | HotpotQA | TriviaQA | StrategyQA |
> |  ----  | ----  |----  | ----  | ----  |
> | Llama2|45.5 | 41.3 | 69.6 | 62.4|
> | RAG | 46.3 | 42.1 | 70.3 | 62.9 |
> | REACT |46.8 | 43.2 | 70.7 | 64.1 |
> | CAD | 47.2 | 43.1| 70.5 | 64.0|
> | RHO| 47.6 |42.9 | 71.1| 63.8|
> | EATQA | 49.1| 44.9 |73.4 | 65.2|
> It shows the significant effectiveness of our evidence-enhanced triplet generation paradigm across diverse datasets.
>
>
> # As for computational costs
> Considering the length of evidences is **much less than the document (about 10% of the document length)**, and the transformer computation cost are **quadratic relation to the input length**, our evidence enhanced triplet generation paradigm **will not significantly increase the computation cost**. In practice, the baseline llama2 finetuning costs about 5 hours and our method costs about 7 hours with one A100 gpu. Considering our significant improvement over informative evidence generation as well as faithful answer reasoning, it shows the effectiveness of our evidence enhanced triplet generation paradigm.

---

### Official Review · Reviewer_edPa · 2024-11-05

**Soundness:** 2
**Presentation:** 2
**Contribution:** 2
**Rating:** 3
**Confidence:** 4

**Summary:**

The paper proposes EATQA (Evidence-Enhanced Triplet Generation Framework), designed to reduce hallucinations in Generative Question Answering (GQA). EATQA leverages a structured approach by generating triplets of Question, Evidence, and Answer (QEA) and using these to reinforce logical consistency. The model is trained on three main tasks: evidence generation, question answering, and query restoration, which improve the alignment between evidence and answers. Tested on MultiRC and QASPER datasets, EATQA achieves state-of-the-art results, effectively reducing hallucination and enhancing answer fidelity by distilling knowledge directly from evidence during inference.

**Strengths:**

1. The paper presents a comprehensive methodology and demonstrates a strong experimental setup. EATQA's effectiveness is validated across two benchmarks, MultiRC and QASPER, where it outperforms prior state-of-the-art models. The paper provides detailed comparisons with competitive LLMs, proving the reliability and effectiveness of the proposed method. Ablation studies further establish the significance of each component in the framework, such as the impact of removing evidence generation or query restoration on performance.
2. The authors provide a clear exposition of EATQA’s architecture and its underlying principles. The paper is well-organized, with clear definitions of the three primary tasks (evidence generation, question answering, and question restoration). Figures, such as the model overview and template instructions, aid in visualizing the complex relationships within the triplet generation framework. Additionally, the equations and methodological breakdown make it accessible to readers familiar with GQA and hallucination mitigation research.

**Weaknesses:**

1. Limited innovation: The paper's proposed three training losses lack technical depth, and this multi-task approach has already been proposed and used in many scenarios. Although there are improvements on two benchmarks, the method does not provide new insights or thoughts for the readers.
2. Insufficient baseline models: The discussion of baseline models for retrieval-enhanced methods in the paper is not comprehensive enough.
3. Limited generalizability: The paper does not conduct experiments on a broader range of datasets, making it difficult to demonstrate the method's generalizability, especially in scenarios where large models are fine-tuned, such as in different types of multi-hop QA scenarios like NQ, TQ, StrategyQA, and MusiQA.
4. Non-standard writing format: There are many citation format errors, images are not in vector format, and there are issues with the image formatting.

**Questions:**

See the weaknesses section.

---

> ### Author Response · Authors · 2024-11-21
> **Response for reviewer edPa**
>
> Thank you very much for providing such insightful and valuable suggestions. We have carefully considered the questions you proposed and would like to respond to each point in detail.
> # As for our innovation and multitask mechanism
> We need to clarify that our innovation **does not lie in the multitask training mechanism and we choose multitask learning because it applies to our training design and theorical analysis.** Specifically, the original reasoning process is hard to verify its correctness because of its containing sentences beyond the surface content of the document. For example in Figure 1, the reasoning process ‘’from year 2002 to year 2006’ is not existed in the original document which brings difficulty to verify its factuality based on the document. Therefore, we **decompose the reasoning ability into 2 phrases: evidence sentence generation and information integration**. In evidence generation, the model is instructed to generate the supporting evidence composing multiple (sub)sentences from the original document and does not conduct information integration. So we can easily discriminate its factuality based on surface text match against the original document. In information integration phrase, the model merges different parts from evidence to derive the answer.
>
> We **tackle two challenges encountered in the two phrases: 1. The evidence sentences are not provided in the original training dataset. 2. The lack of understanding of logical relation among query, evidences and answer**. Some surface similar sentences instead of logical supporting sentences may be mistaken by model as the evidence.
>
> **For the first challenge. We propose the self-reasoning method, which composes candidate generation and correctness verify.** In candidate generation, the LLM is instructed to generate the candidate evidence including the original text from document and out-of-document candidates are filtered to maintain the factuality. In correctness verify, LLM needs to answer the query based on the vanilla generated candidates. The evidences which did not contain the needed information will induce incorrect answers, so we compare the predicted answer against the golden answer to filter the factual faithful but not informative evidences. We name the faithful and informative evidence as correct evidence.
>
> **For the second challenge, we propose two insights**: 1. only correct and informative evidences contain the enough information to recover the original query. 2. Predicted Answer distribution based on correct evidences, and original document should be close. For example in Figure 1, the correct evidence sent 13 contains the important pattern “testing the aircraft in 2006” of the question which is crucial for the query recovery. And the incorrect evidence sent 14 recovers the incorrect query. Based on these insights, we propose our evidence enhanced triplet generation framework and enhance the logical relation among the three parts. By Eq. 1 and 7, we provide the theory analysis of the designed training objective, and unify the different modules in the multi-task framework based on the analysis:
>
> $$ \log\mathbf{P_M}(a |q,e,d) \propto  \log(\mathbf{P_M}(a|d,q)) + \log( \mathbf{P_M}(e|a,d)  ) + \log(\mathbf{P_M}(q|e,a,d)) $$
> $$ \log(\mathbf{P_M} (e,q,d)) \geq  E_{q(a|e,q)} \log(\mathbf{P_M} (e|a,q)) - KL(\mathbf{P_M} (a,d,q) || q(a|e,q))$$ where $\log(P_M)$ is corresponding to our cross-entropy loss.
>
> **The multitask paradigm is only the implementation method which complies to our analysis but not the innovation itself**.

---

> ### Author Response · Authors · 2024-11-21
> **Response for reviewer edPa**
>
> # As for our baselines
> We are sorry for any confusion caused in your understanding about baselines. We compare our methods with 3 existing well-known hallucination mitigation methods [1], RAG [2], CAD [3], and RHO [4]. These baselines are **representative methods of 3 different categories of hallucination mitigation: retrieval Augmented Generation, Introducing New Decoding Strategy, and Utilization of Knowledge Graph (KG)**.
>
> RAG explores a general-purpose fine-tuning recipe for knowledge intensive tasks which combine pre-trained parametric and non-parametric memory for language generation.
> CAD proposes context-aware decoding which follows a contrastive output distribution that amplifies the difference between the output probabilities when a model is used with and without context.
> RHO proposes local knowledge grounding to combine textual embeddings with the corresponding KG embeddings, and global knowledge grounding to equip with multi-hop reasoning abilities.
>
> Our proposed evidence enhanced triplet generation paradigm differ from existing methods in several folds.
>
> 1.	The external information incorporated by existing baselines may be **surface relevant but does not contain the information to support query answering, which introduces distraction for model generation**. For example in Figure 1, the sentence “The Army began developing the Osprey in 1985” is semantically similar as the query but it do not contain the needed information for answering. However, **our self-reasoning stage keep the faithful and informative evidence for training. In training, different modules improve each other with positive correlation from Figure 3, where our ability of generating informative evidences and conducting query reasoning improve as training proceeding**.
>
> 2.	In existing baselines, **the correctly exploit of external information beyond the internal knowledge to solve the query of model remains a challenge**. Considering LLM may **resort to internal knowledge and ignores the original document**, they generate incorrect answers which is not faithful to document** from Table 3. However, in our paradigm, the model needs to **generate the evidence sentence from the document instead of internal knowledge**, so it is trained to focus more on the document which mitigates hallucination. On the other hand, our triplet generation paradigm **enhances the logical relation among evidence, query and answer. The model resort to the faithful and informative evidences instead of internal hallucination**.
>
>
> 3.	Our method **does not need external pretrained retriever or well-designed knowledge base** to enhance the reading comprehension ability of backbone model. Instead, we explore and improve the ability of information retrieval and conducting reasoning of vanilla backbone. This enables our model to **apply to generalized domain or dataset where well designed retriever and KG is hard to derive**.
>
> 4.	We provide the **theory analysis to explain and demonstrate the effectiveness** of our method design. The different objectives **enhance each other in our designed training procedure based on Bayes formulation and probability induction**.
>
> # As for our datasets
> We conducted experiments on two challenging and widespread benchmark multi-hop datasets. To demonstrate our generalization, we conduct experiments on a wide range of multi-hop datasets including NQ [5], TQA [6], StrategyQA [7], and HotpotQA [8] as following with token-level F1 for NQ, TQA and HotpotQA, as well as accuracy for StrategyQA as the evaluation metric. Following React [9], we use 2000 samples as the training set.
>
> | model | NQ  | HotpotQA | TriviaQA | StrategyQA |
> |  ----  | ----  |----  | ----  | ----  |
> | Llama2|45.5 | 41.3 | 69.6 | 62.4|
> | RAG | 46.3 | 42.1 | 70.3 | 62.9 |
> | REACT |46.8 | 43.2 | 70.7 | 64.1 |
> | CAD | 47.2 | 43.1| 70.5 | 64.0|
> | RHO| 47.6 |42.9 | 71.1| 63.8|
> | EATQA | 49.1| 44.9 |73.4 | 65.2|
> It show the significant effectiveness of our evidence-enhanced triplet generation paradigm across diverse datasets.
>
> [1] A Comprehensive Survey of Hallucination Mitigation Techniques in Large
> Language Models. Arxiv 2024.
>
> [2] Retrieval-augmented generation for knowledge-intensive nlp tasks. Arxiv 2021.
>
> [3] Trusting your evidence: Hallucinate less with context-aware decoding NAACL 2024.
>
> [4] RHO:Reducing hallucination in open-domain dialogues with knowledge grounding. ACL 2023.
>
> [5] Natural Questions: A Benchmark for Question Answering Research. TACL 2019.
>
> [6] TriviaQA: A Large Scale Distantly Supervised Challenge Dataset for Reading Comprehension. ACL 2017.
>
> [7] Did Aristotle Use a Laptop? A Question Answering Benchmark with Implicit Reasoning Strategies. TACL 2021.
>
> [8] HotpotQA: A Dataset for Diverse, Explainable Multi-hop Question Answering. EMNLP 2018.
>
> [9] REACT: SYNERGIZING REASONING AND ACTING IN LANGUAGE MODELS. ICLR 2023.

---

> > ### Comment · Reviewer_edPa · 2024-11-29
> > **Response to Authors**
> >
> > Thank you for the author's response. However, it does not address my concerns regarding limited innovation. I will maintain my current rating.

---

### Author Response · Authors · 2024-11-27
**General response and hope for your replies**

Dear Reviewers,

We sincerely thank you for your thoughtful and constructive feedback. We have dedicated considerable time to crafting this rebuttal, as well as updated our paper with the revisions and additional experimental results in the revised PDF. The specific areas addressed include:

1.	Generalization on a broader range of datasets (included in section 5.6 and Table 6.)

2.	Discussion of baselines (included in section 2 and 4.3)

3.	Evaluation of hallucination mitigation (included in section 5.3 and 5.4)

4.	Reference evidences (included in section 3.2)

5.	Computation cost (included in section 5.8)

Once again, we are grateful for your valuable insights, which have significantly enhanced our work. We have made every effort to comprehensively address all concerns.

We hope to receive your replies and address any concerns in detail.

Sincerely,

---

### Meta-Review · Area_Chair_FEtD · 2024-12-20

**Metareview:**

The paper introduces a method designed to alleviate hallucinations in Generative Question Answering by employing a structured approach. This involves the creation of triplets consisting of a Question, Evidence, and Answer, which are used to enhance logical consistency.

While the authors have addressed several points raised during the review process, significant concerns remain unresolved after the rebuttal:

- Limited novelty and technical depth: The proposed contributions rely on training loss functions and multi-task learning approaches. These techniques has been proposed and applied in various scenarios. As such, the method lacks fresh insights or contributions that would advance the field.

- Clarity and writing issues: Two reviewers noted that the paper is partially difficult to follow. More critically, it contains errors in writing format, including improper citation formatting and image formatting.

Given these major unresolved issues, we all agree that the submission does not currently meet the standards required for acceptance at ICLR. We hope the feedbacks provided will help the authors improve the paper in future revisions.

**Additional Comments On Reviewer Discussion:**

No changes after rebuttal. The unsolved points have been included in the meta-review.

---

### Decision · Program_Chairs · 2025-01-22

Reject